

# ONE3A: one-against-all authentication model for smartphone using GAN network and optimization techniques

Mohamed Meselhy Eltoukhy[1], Tarek Gaber[2,3], Abdulwahab Ali Almazroi[1] and Marwa F. Mohamed[3]

[1] Department of Information Technology, College of Computing and Information Technology at Khulais, University of Jeddah, Jeddah, Saudi Arabia
[2] School of Science, Engineering, and Environment, University of Salford, Salford, United Kingdom
[3] Department of Computer Science, Faculty of Computers and Informatics, Suez Canal University, Ismailia, Egypt

Corresponding author
Mohamed Meselhy Eltoukhy,
mmeltoukhy@uj.edu.sa

## ABSTRACT

This study focuses on addressing computational limits in smartphones by proposing an efficient authentication model that enables implicit authentication without requiring additional hardware and incurring less computational cost. The research explores various wrapper feature selection strategies and classifiers to enhance authentication accuracy while considering smartphone limitations such as hardware constraints, battery life, and memory size. However, the available dataset is small; thus, it cannot support a general conclusion. In this article, a novel implicit authentication model for smartphone users is proposed to address the one-against-all classification problem in smartphone authentication. This model depends on the integration of the conditional tabular generative adversarial network (CTGAN) to generate synthetic data to address the imbalanced dataset and a new proposed feature selection technique based on the Whale Optimization Algorithm (WOA). The model was evaluated using a public dataset (RHU touch mobile keystroke dataset), and the results showed that the WOA with the random forest (RF) classifier achieved the best reduction rate compared to the Harris Hawks Optimization (HHO) algorithm. Additionally, its classification accuracy was found to be the best in mobile user authentication from their touch behavior data. WOA-RF achieved an average accuracy of 99.62 ± 0.40% with a reduction rate averaging 87.85% across ten users, demonstrating its effectiveness in smartphone authentication.

## INTRODUCTION

Smartphones have become a fundamental aspect of individuals' personal and professional lives (*Van Deursen et al., 2015*). Mobile phones are no longer limited to two-way conversations. They serve as digital cameras, navigation tools, web browsers, personalized assistants, and host various useful applications, including mobile banking and e-commerce. Consequently, numerous applications, such as social networks, online gambling, emails,

and electronic payments, are at risk. For instance, the European Union implemented the online Payment Services Directive (PSD2—DSP2) as of September 14, 2019. The directive aims to enhance security while providing customers with practical options. This inclination is pervasive across all digital services (*Gernot & Rosenberger, 2024*).

The security options provided by the hardware and operating systems of smartphone devices play a crucial role in users' choices for securing them. Common security measures include PINs, passwords, fingerprints, and facial recognition. However, these static authentication methods are susceptible to various attacks such as side-channel, dictionary, guessing, and spoofing attacks. Notably, the device remains vulnerable to unauthorized access when unlocked, posing a potential risk to all stored data (*Cariello et al., 2024*). To address this vulnerability, continuous authentication is recommended.

Research conducted in recent years has explored the feasibility of continuous identity authentication. Continuous identity authentication verifies users' identities throughout the session activity, actively identifying abnormal or harmful behavior. This approach complements traditional static authentication methods, enhancing overall system security. Behavioral aspects, particularly keystroke dynamics, serve as a means of continuous identity authentication. Keystroke dynamics analyze patterns such as time and rhythm, uniquely identifying individual users based on their keyboard interaction (*Yang et al., 2023*). Importantly, the implementation of keystroke dynamics verification systems is cost-effective as it requires no additional hardware (*Stragapede et al., 2023*). Several studies (*Yang et al., 2023*; *El-Soud et al., 2021*; *Almazroi & Eltoukhy, 2023*; *Progonov, Prokhorchuk & Oliynyk, 2020*) have proposed behavioral biometrics-based authentication for continuous smartphone users.

Feature selection is employed to discern significant features from inconsequential features within a predetermined feature collection (*Hamed & Mohamed, 2023*; *Hamed & Nassar, 2021*; *Ablel-Rheem et al., 2020*). Feature selection aims to minimize the size of high-dimensional classification issues while improving prediction accuracy in classification problems (*Sağbaş & Ballı, 2024*). In practical scenarios, data representation often encompasses numerous aspects, some of which may be non-essential. This leads to a scenario in which certain characteristics assume the roles of others. Conversely, the crucial characteristics have a significant impact on the outcome, providing valuable information that might be diminished if any element is eliminated (*Almomani, 2020*). Determining the essential set of characteristics is a challenging and computationally expensive undertaking. Metaheuristics have emerged as a powerful and reliable approach for handling a wide range of optimization issues in recent years (*Tahoun et al., 2020*). Metaheuristics have demonstrated superior performance compared to existing algorithms due to their ability to bypass the need for analyzing the entire search space.

Mobile continuous authentication models typically employ feature selection to enhance the accuracy of machine learning-based biometric authentication for smartphone users (*Hamed & Nassar, 2021*). The literature also emphasizes the use of bioinspired feature extraction algorithms such as Grey Wolf Optimization (GWO) (*Almazroi & Eltoukhy, 2023*), Particle Swarm Optimization (PSO) (*Rostami et al., 2020*), Whale Optimization Algorithm (WOA) (*Mirjalili & Lewis, 2016*), Harris Hawks Optimization (HHO)

(*Heidari et al., 2019*), and Bayesian Optimization Algorithm (BOA) (*Yang, Liu & Wen, 2024*). However, to the authors' knowledge, bioinspired feature selection for the one-against-all classification problem in smartphone authentication has not been addressed.

This study aims to address the computational limits of smartphones and proposes an efficient authentication mechanism that offers implicit authentication for smartphone users without requiring additional hardware. To achieve this goal, we explored various wrapper feature selection strategies and classifiers that could enhance authentication accuracy while taking into consideration the computational limitations of smartphones, including limited hardware, battery life, and memory size. The investigation utilized a public dataset on user behavior (*El-Abed, Dafer & El Khayat, 2014*), specifically focusing on keystrokes of touch screens. The following is a list of the principal contributions of this work:

- Proposing a novel implicit authentication model for smartphone users, addressing the one-against-all classification problem in smartphone authentication. This was achieved using the CTGAN algorithm and a newly proposed WOA-based feature selection technique. To the best of the author's knowledge, this approach has not been previously employed within the research field of mobile authentication.
- Thoroughly evaluating the efficacy of two wrapper feature selection strategies (WOA and HHO) on the implicit authentication method's performance. This was done using different types of performance metrics to ensure the effectiveness of this model. These metrics include accuracy, precision, recall, F-measure, and Kappa. Using these metrics, it was shown that our proposed model was comprehensively evaluated from different aspects to prove its effectiveness and quality.
- Using a conditional tabular generative adversarial network (CTGAN) to generate synthetic data based on actual data. To overcome the imbalance problem between selected users' data and other users' data and to augment the small size of the original data. The quality of the CTGAN-generated data was evaluated using several metrics: Quality Score, Column Shapes, and Column Pair Trends of the generated data.

## RELATED WORK

Over the past several decades, smartphone authentication has predominantly relied on a knowledge-based authentication mechanism, wherein users are required to possess knowledge (*e.g.*, password) about a certain entity. Nevertheless, scholarly investigations have indicated that the implementation of this approach on mobile devices presents several barriers, including inadequate security measures and a deficiency in user-friendliness (*Stylios et al., 2021*). Numerous research has been undertaken in recent years to address these barriers. This section will provide an overview of the relevant literature on smartphone authentication. The focus will be placed on the use of bioinspired algorithms in the selection of optimal characteristics for mobile authentication techniques, and the subsequent enhancements in time complexity and authentication accuracy achieved through their implementation.

## Behavioral biometrics-based authentication for smartphone users

*Stylios et al. (2016)* conducted a study on continuous authentication and behavioral biometrics systems specifically designed for mobile devices. They provide a classification of behavioral biometric approaches and offer a comprehensive explanation of the authentication process for mobile devices. The literature was critically discussed, and a summary of the lessons learned and research difficulties was presented. Using different machine learning algorithms, *Nader et al. (2015)* proposed a fusion authentication technique that incorporates two forms of authentication, namely implicit authentication and continuous authentication, in response to the growing tendency to safeguard smartphones through user authentication. For the experiment, a range of features were extracted from the interactions that participants had with Android cellphones. The findings indicated that PSO-Radial Basis Function Network (RBFN) offers superior performance in the context of user authentication. PSO-RBFN yields an average error rate (ER) of 1.9%.

In their study, the authors of *El-Soud et al. (2021)* introduced a novel approach to mobile phone authentication, utilising an implicit authentication method. They have also developed the methodology in such a way that it does not require any further expenditure on supplementary hardware resources. The researchers reached the conclusion that employing a filter-based strategy represents the most effective approach for extracting features in an inferred authentication system.

The work proposed in *Tharwat et al. (2019)* utilizes mobile-based touch mobile phone keyboard dynamics to address the challenge of personal identity. The suggested method comprises two primary stages: feature selection and categorization. The genetic algorithm (GA) is employed to select the most significant features. Additionally, the Bagging classifier uses the chosen features to identify individuals by comparing the features of the unknown individual with the labeled features. The final decision is made by fusing the outputs of each Bagging classifier. The work proposed in *Almazroi & Eltoukhy (2023)* automatically authenticates users based on their touch behavior by combining a random forest (RF) classifier with the GWO, which is used as a feature selection strategy. The initial step involves selecting the most important features, and the random forest classifier is then utilized to determine which user is using the smartphone. Their method was evaluated using a publicly accessible benchmark dataset from RHU (*El-Abed, Dafer & El Khayat, 2014*), resulting in an accuracy of 97.89%.

In *Yang et al. (2023)*, the authors presented dual attention networks with pre-trained models for content and keystrokes for continuous authentication. The model considers the user's entry of "text" while pressing keys as a valuable asset, in addition to the more traditional aspects of keystroke dynamics. Specifically, it captures textual aspects using the popular pre-trained model named Robustly Optimized BERT Pretraining (RoBERTa). Next, it runs both traditional and textual features through their suggested dual attention networks. These features are combined by their networks to obtain final representations. The model is tested using two datasets named Clarkson II (*Murphy et al., 2017*) and Buffalo (*Sun, Ceker & Upadhyaya, 2016*).

The work proposed in *Al-Saraireh & AlJa'afreh (2023)* integrates free text-based keystroke dynamics (KD) or swipe dynamics (SD) to propose an enhanced smartphone

**Table 1  A summary of the related work.**

| Study | # users | Feature selection - classifier | Dataset | Performance |
|---|---|---|---|---|
| *Nader et al. (2015)* | 20 | PSO - RBFN | Original Dataset | Avg. ER = 1.9% |
| *El-Soud et al. (2021)* | 51 | Rank - RF | RHU *El-Abed, Dafer & El Khayat (2014)* | Acc. = 97.80% |
| *Tharwat et al. (2019)* | 51 | GA - Bagging | RHU *El-Abed, Dafer & El Khayat (2014)* | Acc. = 83.8% |
| *Almazroi & Eltoukhy (2023)* | 51 | GWO - RF | RHU *El-Abed, Dafer & El Khayat (2014)* | Acc. = 97.89% |
| *Yang et al. (2023)* | 103 | RoBERTa - CKDAN | Clarkson II *Murphy et al. (2017)* | EER = 6.47% |
| | 148 | | Buffalo *Sun, Ceker & Upadhyaya (2016)* | EER = 3.49% |
| *Al-Saraireh & AlJa'afreh (2023)* | 56 | CBA and IR - RF | BB-MAS *Belman et al. (2019)* | Acc. = 99.98% |
| *Tse & Hung (2022)* | 31 | RNN / late fusion | Private Dataset | Acc. = 95.29% |
| *AbdelRaouf et al. (2023)* | 51 | CNN - CatBoost | CMU *Killourhy & Maxion (2009)* | Acc. = 99.95% |

continuous authentication model. The KD and SD raw data are initially obtained by the model from the BB-MAS dataset (*Belman et al., 2019*), respectively. Appropriate pre-processing and feature extraction techniques are then applied. To obtain the optimal subset feature form KD raw data, the most effective features are chosen utilizing the correlation-based analysis (CBA) feature selection approach. Conversely, the identical procedures are applied for the SD raw data, with the exception of the feature selection method, which employs the importance ranking (IR) feature selection approach as it yields the best outcomes. To test the feature-level fusion stage, the selected features from the KD and SD are concatenated and applied to the RF classifier. The model's performance showed an accuracy of 99.98% and the lowest equal error rate (EER) rate of 0.02% in multi-class classification.

*Tse & Hung (2022)* established and incorporated a trajectory model for keystroke dynamics into the behavioral biometric model, in addition to using spatial, temporal, and swiping features. The trajectory model utilized more information connected to the keystroke behavioral pattern than other models, making it harder to pretend to be a genuine user. A weighted product rule, in conjunction with a late fusion method, was used to merge all four feature categories. The final classification outcome was obtained by combining the classifier's outputs for each feature. This was achieved by independently feeding each of the four feature models into different recurrent neural networks (RNNs). The results of the experiment showed that biometric models that employ the trajectory model have higher security because they incorporate more elements that describe user behavior.

*AbdelRaouf et al. (2023)* developed an optimized convolutional neural network that utilizes quantile transformation and data synthesis to extract improved features and enhance accuracy. They proposed an approach for the training and testing stages that also employs ensemble learning techniques. Additionally, the model was evaluated using a publicly accessible benchmark dataset from Carnegie Mellon University (CMU) (*Killourhy & Maxion, 2009*), resulting in an average accuracy of 99.95%, an EER of 0.65%, and an area under the curve(AUC) of 99.99%. Table 1 summarizes the related work.

## Bio-inspired-based methods for feature selection

Various bioinspired algorithms have been utilized for solving the feature selection problem in many applications. The authors in reference (*Hraiba, Touil & Mousrij, 2020*) introduced an enhanced version of the binary grey wolf optimizer (IGWO) as a means of improving the feature selection technique. The outcomes of the study indicate that the IGWO algorithm possesses the ability to conduct global searches effectively and efficiently. Consequently, it is deemed appropriate for the purpose of conducting reliability analysis in the field of engineering. In their work, *Chantar et al. (2020)* introduced an improved version of the binary grey wolf optimizer which was then used to test its impact on feature selection using various machine learning approaches, including decision trees (DT), K-nearest neighbour (KNN), naive Bayes (NB), and support vector machine (SVM) classifiers. The findings demonstrate a notable improvement in feature selection when employing the improved grey wolf optimizer in conjunction with the SVM classifier. In their study, *Tahoun et al. (2020)* employed the grey wolf optimizer as feature selection technique for features extracted from wavelet and curvelet sub-bands in order to facilitate mammography classification. The findings indicate that the utilization of binary grey wolf optimization effectively extracts the most optimal characteristics.

*Moradi & Gholampour (2016)* proposed a novel approach for addressing the feature selection problem by integrating filter and wrapper techniques. Their method utilizes a PSO algorithm to achieve efficient feature selection aiming to locate the optimal subset of features that are both non-redundant and statistically significant. The strategy employed in their study is based on using PSO to address numerical and medicinal applications using a single-objective approach. Furthermore, *Rostami et al. (2020)* integrated graph theory and PSO as a means of solving the feature selection problem inside medical applications, with the aim of enhancing diagnostic accuracy. The researchers employed the node centrality criterion to provide a novel approach for initializing the particles of the PSO algorithm. By employing a multi-objective fitness function, the researchers successfully partitioned the characteristics into two distinct sets. The first set comprises features that exhibit the least similarity to each other, while the second set consists of traits that are most pertinent to the target class. By utilizing these two distinct categories of features, it is possible to effectively diagnose an illness. The authors of *Al-Tashi et al. (2019)* suggest using a combination of two bioinspired optimization techniques, binary grey wolf and particle swarm optimization, for feature selection. The results of the experiments suggest that the combined algorithm performed better than the alternative strategies, specifically the binary grey wolf optimization, the binary particle swarm optimization, and the binary genetic algorithm.

In *Sharawi, Zawbaa & Emary (2017)*, the researchers used WOA algorithm for wrapper-based feature selection approach. Using the classification accuracy as a single objective function, the proposed approach was able to determine the optimal feature subset for classification accuracy. This strategy was utilized to keep as few features as possible from the data set while maintaining the highest level of accuracy feasible. In *Nematzadeh et al. (2019)*, the authors proposed wrapper-based feature selection method using the WOA algorithm and the simulated annealing technique is created for the purpose of feature

selection and optimization. Incorporating the simulated annealing helped in improving the exploitation in the feature selection process by looking for the most promising regions identified by the WOA algorithm. On high-dimensional medical datasets, *Nematzadeh et al. (2019)* developed a frequency-based filter feature selection approach based on the whale algorithm. The WOA is used in conjunction with a filter criterion to eliminate the features that aren't significant in this procedure. After then, the reminder features are prioritized based on another filtering process, called mutual congestion, which is used to rank the features. In *Awad, Ali & Gaber (2020)*, butterfly optimization algorithm (BOA) was combined with chaotic maps to build a new feature selection technique (CBOA) to enhance diversity and mitigate the risk of being trapped in local minima. The performance evaluation of the proposed CBOA was carried out, pitting CBOA against six other meta-heuristic algorithms. The findings indicate that the utilization of chaotic maps in the standard BOA can enhance its performance, resulting in increased accuracy by a significant margin.

From the given literature above, a few observations can be noticed. A range of mobile device authentication approaches have been suggested, with a notable focus on the use of machine learning algorithms and biometric characteristics for the purpose of smartphone users' authentication. In addition, the literature has emphasised the significance of employing bioinspired algorithms (GWO, PSO, WOA, and BOA) for the purpose of feature selection. However, there was no work addressing the feature selection using bioinspired techniques for the one-against-all classification problem in the smartphone authentication. To the best of the author's knowledge, this approach has not been previously employed within the research field of mobile authentication.

## ONE-AGAINST-ALL AUTHENTICATION (ONE3A) MODEL

The proposed model, ONE3A, given in Algorithm 1, aims to determine whether a user is legitimate or illegitimate for smartphones. It offers an effective authentication approach, providing implicit authentication for smartphone users while avoiding additional costs associated with specialized hardware and addressing the computational limits of smartphones. As depicted in Fig. 1, it consists of four steps: 1-data pre-processing, 2- data splitting, generation, and random selection of illegitimate users, 3- feature selection, and 4- finally, a classification step that is used to determine whether the user is legitimate or illegitimate. These steps are discussed and explained in detail in the following subsections.

### Data pre-processing

Data pre-processing is the initial stage of machine learning, during which the data is encoded to put it in a format that allows the computer to analyze or understand it rapidly. The most significant factor influencing classfication algorithm's ability to perform well in generalization is data pre-processing. Pre-processing is crucial to model construction; estimates suggest that it can account for 50% to 80% of the whole classification process. Enhancing the quality of the data is also necessary for increased performance (*Maharana, Mondal & Nemade, 2022*). Therefore, datasets must undergo essential data pre-processing

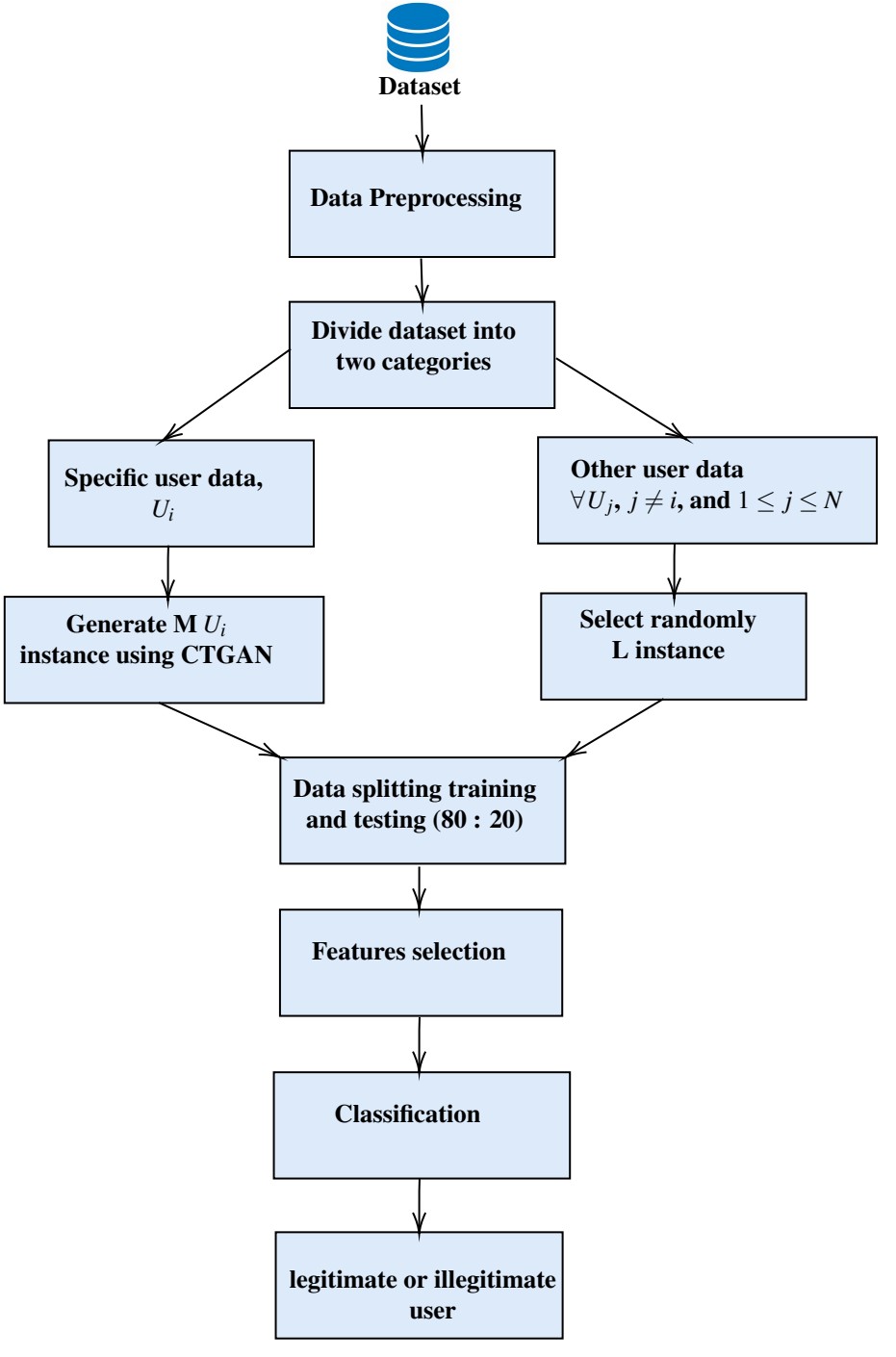

**Figure 1** The architecture of the proposed (ONE3A) model.

---

**Algorithm 1 One-against-all authentication algorithms (ONE3A)**

---

**Input:** $U$ // Authentication dataset

**Output:** legitimate or illegitimate user

  1: load dataset $(U)$

  2: Data pre-processing (e.g., remove null values)

  3: Split data into two classes legitimate user (LU) and illegitimate users (IU)

  4: $U_{lu}$ = Generate $M$ instance for (LU) using CTGAN

  5: $U_{iu}$ = Select randomly $L$ instance from (IU) data

  6: Merge the data of $U_{lu}$ and $U_{iu}$ in to one dataset $U$

  7: Split data into train ($Utrain$) and test ($Utest$)

  8: Perform the feature selection using optimization algorithms

  9: Build the classification model using selected features

 10: Validate and test the generated model

---

steps, including the removal of null values. In optimization and machine learning models, the presence of null values can produce issues, such as diminished prediction accuracy or model failure (*Hosea, 2021*).

## Data splitting, generation and selection

Data splitting is a crucial step in the proposed algorithm, involving the division of the dataset into two classes: legitimate or illegitimate user. Let $N$ represent the total number of users, where $N$ equals 51. The dataset is then divided into two classes, denoted as $C_1$ and $C_2$. The first class, denoted as $C_1$, comprises a specific user data represented by $U_i$, where the specific user is $U_1$ with a sample count ($l_1$) of 18. The second class, $C_2$, includes data from other users, denoted as $U_j$, where $j$ ranges from 2 to 51. The sample count for this class ($l_2$) is calculated as $954 - 18 = 936$, where 936 is the number of samples for all users. It is evident that an imbalance exists between $l_1$ and $l_2$. Consequently, data generation is applied to $C_1$, while data selection is applied to $C_2$ as follow:

$$\begin{cases} U_i, & 1 \le i \le N \\ \forall U_j, & j \ne i \text{ and } 1 \le j \le N. \end{cases} \tag{1}$$

Generative adversarial network (GAN) with the capability to generate synthetic data based on actual data. One of the advantages of using GAN is its ability to create synthetic data while preserving the correlations between various columns in the real data. Another benefit is its quick data generation, as the time complexity is not dependent on the number of rows in the real data. Additionally, its simplicity of operation, with the ability to be turned on and off with a single input, adds to its ease of use (*Xu et al., 2019*).

In this study, CTGAN is utilized for generating $C_1$ data 20 times. Consequently, the number of samples in $C_1$ ($L_1$) becomes 21 times the original samples. For example, if the number of samples for $U_1$ ($L_1$) is 18, the total number of generated data samples is $18 \times 20 = 360$, plus the number of actual data samples (18), resulting in a total of 378 samples.

$$M(\text{generated data}) = 20 * l_1 \tag{2}$$

**Table 2    Data splitting, generation and selection example.**

| | Class 1 ($C_1$) | | | Class 2 ($C_2$) | Used Dataset |
|---|---|---|---|---|---|
| | Data generation | | | Randomly selection | Total sample |
| $U_i$ | $l_1$ | $M$ | $L$ | $L$ | 2*L |
| 1 | 18 | 360 | 378 | 378 | 756 |
| 6 | 21 | 420 | 441 | 441 | 882 |
| 8 | 15 | 300 | 315 | 315 | 630 |
| 9 | 20 | 400 | 420 | 420 | 840 |
| 11 | 20 | 400 | 420 | 420 | 840 |
| 12 | 16 | 320 | 336 | 336 | 672 |
| 14 | 22 | 440 | 462 | 462 | 924 |
| 26 | 19 | 380 | 399 | 399 | 798 |
| 46 | 17 | 340 | 357 | 357 | 714 |
| 49 | 18 | 360 | 378 | 378 | 756 |

$$L = M(\text{generated data}) + l_1(\text{actual data}) \tag{3}$$

$$L = 21 * l_1. \tag{4}$$

For fairness, we randomly select $L$ samples from class two ($C_2$) samples ($l_2$). Therefore, the total length for the used dataset is equal to $2 \times L$. Table 2 shows examples for data from 10 users. It is important to note that each row in Table 2 represents an independent experiment from the other rows. We apply the following steps (*e.g.*, feature selection, classification, and performance evaluation) to each row separately.

## Feature selection

In the data classification process, feature selection plays a crucial role. Given that datasets often include a multitude of features, not all of them are necessarily significant. Failure to address this can adversely affect the classification procedure. Therefore, feature selection becomes a vital stage in establishing an effective system, as emphasized in prior research (*El-Soud et al., 2021*). The objective of this research is to devise a reliable smartphone user authentication method. Unnecessary features in this authentication method can impact its accuracy rate and/or processing time, underscoring the importance of thoughtful feature selection.

One of the most intriguing metaheuristic optimization algorithms for feature selection is swarm intelligence (SI), with its conceptual roots dating back to 1993 (*Beni & Wang, 1993*). SI algorithms draw inspiration from the collective behavior observed in natural flocks, colonies, and herds. Noteworthy advantages of SI algorithms include their ability to track information about the search space during iteration, unlike evolutionary algorithms (EAs), which may discard knowledge from previous generations. SI algorithms often retain the most recent optimal solution in memory, and they typically require minor adjustments

to fewer variables. Moreover, SI algorithms are known for their ease of implementation (*Mirjalili, Mirjalili & Lewis, 2014*). In this study, we leverage two prominent SI algorithms, WOA and HHO, both recognized for their effectiveness in feature selection.

The WOA, developed by *Mirjalili & Lewis (2016)*, draws inspiration from the bubble-net feeding behavior observed in humpback whales during foraging. Humpback whales employ a bubble net to catch prey, allowing them to hunt near the surface. In this process, the whales swim in a '6'-shaped manner, creating a net to trap their prey. The WOA mimics this behavior through two key phases: the exploitation phase, characterized by a spiral bubble-net attack tactic to encircle a target, and the exploration phase, involving the search for prey at random (*Mafarja & Mirjalili, 2018*).

The HHO algorithm, introduced by *Heidari et al. (2019)*, is inspired by the cooperative hunting tactics observed in Harris hawks. Known for their exceptional intelligence among avian species, Harris hawks demonstrate sophisticated teamwork in hunting. The HHO algorithm incorporates both local and global searches, enhancing its ability to effectively balance exploitation and exploration search methods (*Shehab et al., 2022*).

Generally, WOA is relatively simple to implement and effective in global search due to its focus on mimicking the efficient hunting behavior of whales. On the other hand, HHO is more complex due to its incorporation of teamwork-inspired search methods. For solving the mobile authentication problem, WOA might be suitable due to its simplicity and effectiveness in global search. However, HHO could also be considered for its ability to balance exploration and exploitation, which could potentially lead to better optimization and adaptation to varying authentication scenarios. The aim of applying WOA and HHO is to determine which one is more efficient in solving the current problem of mobile authentication.

The primary objectives of the employed metaheuristic optimization algorithms (HHO and WOA) are to maximize the rate of feature reduction, denoted as f(fr) Eq. (5), and minimize the classification error f(ce) Eq. (6). However, f(ce) holds greater significance than f(fr). The success of WOA and HHO is contingent on not only decreasing the number of chosen features but also ensuring that the error rate diminishes. To consolidate these objectives into a single metric, referred to as fitness in Eq. (7), f(ce) is assigned the highest weight. In each generation (gen) of HHO and WOA, the fitness value is evaluated. If the fitness value is minimized, the feature list is updated. Conversely, if the fitness value does not decrease, the feature list remains unchanged and is passed on to the next generation.

$$f(fr) = NSF/NF \tag{5}$$

where, NSF is number of selected features and NS is number of all features.

$$f(ce) = 1 - Acc, \tag{6}$$

where, Acc. is a classification accuracy rate as defined in Eq. (8).

$$fitness = w * f(ce) + (1 - w) * f(fr) \tag{7}$$

where w represents a weight, and it's equal 0.99.

## Classification

The classification task in this step is a binary classification, where the first class consists of legitimate user, and the second class comprises illegitimate users. While various ML models can predict authentication, our focus is on three models that demonstrated superior performance KNN, random forest (RF) and support vector machine (SVM). The aim of using three classifiers is to test the performance of three different strategies. The KNN classifier falls under instance-based learning methods, making predictions based on the closest instances in the training data. The random forest approach integrates multiple classifiers through ensemble learning to enhance model performance, while SVM's strategy emphasizes maximizing the margin between classes to improve generalization to unseen data, thereby achieving robust classification performance.

KNN is a popular non-parametric technique used for regression or classification. It leverages inter-sample similarity observed in the training set to intuitively classify unlabeled samples. The key parameter in KNN is the number of neighbors considered. When this value is small, the model's decision boundary becomes complex and prone to overfitting. Conversely, a high number of neighbors results in a simpler decision boundary, which may lead to underfitting. Consequently, selecting an appropriate value for this parameter is crucial for optimal model performance (*Wan et al., 2018*).

RF is a widely used classification and regression method known for its significant success. It operates by aggregating predictions from multiple randomized decision trees, averaging their outputs. RF is versatile and well-suited for large-scale problems. It provides variable importance measures and adapts effectively to diverse learning tasks (*Biau & Scornet, 2016*).

SVM is a well-known supervised machine learning method. It operates by first non-linearly transforming data into higher-dimensional spaces. In the second phase, SVM establishes a linear optimum hyperplane or decision boundary to separate points of different classes (*Aggarwal, 2018*). The objective in SVM is to maximize the margin between the hyperplane and the closest training data points, ensuring effective separation.

## EXPERIMENTAL WORK

In this section, we developed two main experiments and used a publicly available dataset (more details below) in order to assess the ONE3A. The first experiment aims to measure the performance of generated data by CTGAN. While the second experiment aims to measure the performance of ONE3A. Widely used performance metrics (more details below) for user authentication methods are used to quantify the outcomes of two experiments. Mainly, three classifiers (SVM, KNN, and RF) and two optimization algorithms (HHO and WOA) are employed. Each algorithm undergoes five iterations, and the average and standard deviation are recorded for convergence speed, high reduction ability, and performance metrics such as accuracy, recall, precision, and F1 Score, *etc.*

The optimization techniques are implemented using Jx-WFST (Wrapper Feature Selection Toolbox) (*Too, 2021*). The experiments were conducted on an 11th Gen Intel(R) Core(TM) i5-1135G7 @ 2.40 GHz 2.42 GHz Ram 8.00 GB. It's important to note that the

| Table 3 | Algorithms parameters. |
|---|---|
| HHO | No |
| WOA | $b = 1$# constant |
| KNN | $k = 5$ |
| pop | 10 |
| gen | 100 |
| iteration | 5 |

number of generations *gen* is set to 100, and the population *pop* equals 10 for all users. Table 3 summarizes the parameter settings used for each algorithm.

## Experimental setup
### Datset
The dataset employed in this study is the RHU touch mobile keystroke dataset, publicly released in *El-Abed, Dafer & El Khayat (2014)*. This dataset was derived from 51 individuals, each tasked with entering the password "rhu.university" 15 times across three distinct sessions. These sessions were conducted with an average interval of 5 days between each. Notably, data were independently collected for each session, resulting in a dataset typically comprising 955 samples. The participants encompassed individuals of various ages and included both males and females.

For each user, four primary features were extracted from the dataset: PR (time between key pressure and key release), PP (time between two key pressures), RP (time between key release and key pressure), and RR (time between two key releases). Figure 2 explains the timing features (*e.g.*, PR, PP, RP, and RR) for dynamics keystroke. Specifically, RR, RP, and PP each have 13 subfeatures, while PR has 14 subfeatures. Consequently, each user (or class) is characterized by a total of 53 features.

### Performance evaluation metrics
The performance evaluation includes assessing the effectiveness of the proposed model through three different learning strategies,instance-based, ensemble-based and hyperplane. KNN, RF and SVM were selected as examples of each learning strategy. For each classification experiment, a confusion matrix is constructed, detailing the number of cases classified (*e.g.*, legitimate or illegitimate user). Assuming the existence of true positives $T_P$, false positives $F_P$, true negatives $T_N$ and false negatives $F_N$, the correctness of each combination is carefully evaluated. Additionally, various performance metrics are computed, encompassing sensitivity, precision, specificity, accuracy, F1-score, and the Kappa index. These metrics provide a comprehensive analysis of the model's capabilities, enabling researchers to compare multiple models and select the most suitable one for their specific requirements. A detailed breakdown of the model's performance for each case allows researchers to assess its applicability and draw well-informed conclusions about their data.

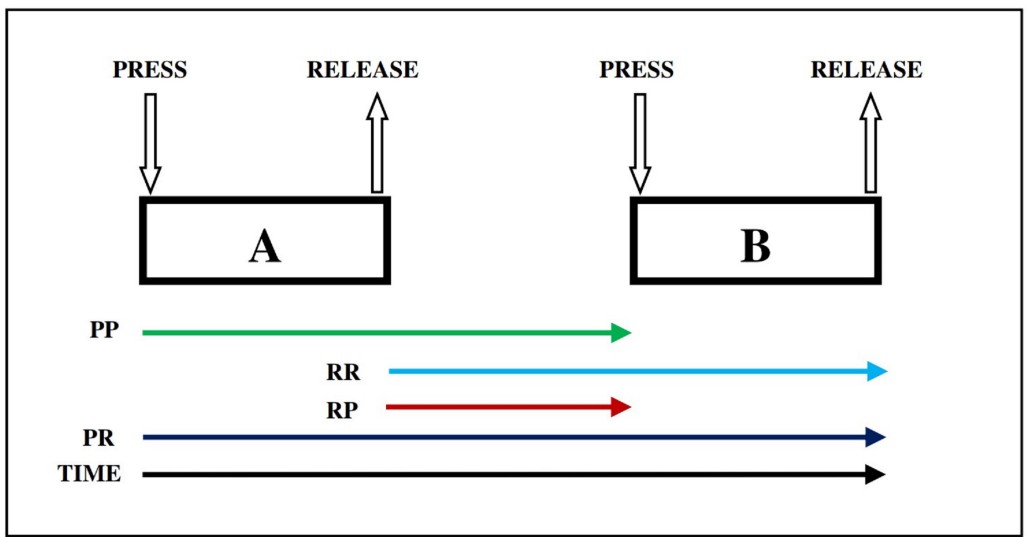

**Figure 2** Dynamics keystroke illustration: press-to-release (PR), press-to-press(PP), release-to-release (RR), and release-to-press (RP) (*Adesina & Oyebola, 2021*).

The accuracy (Acc) is represented by the following equation, which denotes the ratio of the number of correct classifications to all investigated cases.

$$\text{Acc} = \frac{T_P + T_N}{T_P + T_N + F_P + F_N}. \tag{8}$$

The number of successfully classified positive cases was introduced by the sensitivity (recall), which is represented by the following equation.

$$\text{Sensitivity} = \frac{T_P}{T_P + F_N}. \tag{9}$$

The precision is represented by the number of correctly classified cases out of all positive cases classified as given below:

$$\text{Precision} = \frac{T_P}{T_P + F_P}. \tag{10}$$

The number of correctly identified negative cases was introduced by the specificity, which can be stated in the following equation.

$$\text{Specificity} = \frac{T_N}{T_N + F_P}. \tag{11}$$

The harmonic mean of recall and precision was introduced by the F1-score and is represented by the following equation.

$$\text{F1-score} = \frac{2 * \text{Precision} * \text{Recall}}{\text{Precision} + \text{Recall}}. \tag{12}$$

The following formula represents the Kappa index.

$$\text{Kappa} = \frac{2 \times (T_P \times T_N - F_N \times F_P)}{(T_P + F_P) \times (F_P + T_N) + (T_P + F_N) \times (F_N + T_N)}. \tag{13}$$

**Table 4  CTGAN performance.**

| User | N | M | Quality score | Column shapes | Column pair trends |
|------|-----|-----|----------------|----------------|----------------------|
| 1 | 18 | 360 | 89.96% | 94.29% | 85.63% |
| 6 | 21 | 420 | 91.84% | 94.47% | 89.22% |
| 8 | 15 | 300 | 90.92% | 93.84% | 87.99% |
| 9 | 20 | 400 | 89.70% | 94.12% | 85.27% |
| 11 | 20 | 400 | 91.75% | 94.76% | 88.75% |
| 12 | 16 | 320 | 88.37% | 94.14% | 82.59% |
| 14 | 22 | 440 | 90.65% | 94.55% | 86.75% |
| 26 | 19 | 380 | 90.27% | 94.34% | 86.20% |
| 46 | 17 | 340 | 90.26% | 94.03% | 86.49% |
| 49 | 18 | 360 | 90.95% | 94.72% | 87.18% |

## Experiment 1: data generation using CTGAN

In this experiment, CTGAN is employed to generate data for ten users. As indicated in Table 4, each user's data is generated 20 times the number of original records. For instance, if the original number of records for user 1 is 18, then the number of generated records is $20 \times 18 = 360$.

To evaluate the data quality generated by CTGAN, three performance metrics are employed to evaluate the correlations and column shapes of the synthetic data, thereby determining its quality. The first one of these metrics is the quality score, which assesses the quality of synthetic data in terms of its effective creation.

Another metric involves measuring the column shapes and calculating a percentage that reflects the similarity between the value distributions of actual and synthetic data. A higher score for a column indicates that its values are distributed in a manner consistent with the original dataset distribution. These scores for each column are averaged to derive the dataset overall score.

The third metric assesses the trend between two columns, explaining how they change concerning each other, akin to correlation. A higher score indicates more similarity in patterns. As illustrated in Table 4, the quality score, column shapes, and column pair trends of the generated data reach up to 91.84%, 94.76%, and 89.22%, respectively.

## Experiment 2: ONE3A model application

The goal is to compare the performance of two metaheuristic algorithms, WOA and HHO, in selecting the feature set that yields the highest classification performance across three different learning strategies: KNN, RF and SVM. In essence, the objective is to identify the optimal combination of WOA and HHO with each of these classifiers. This investigation aims to determine which algorithm, paired with which classifier, achieves the most effective feature selection for classification tasks.

The produced set of feature from Experiment 1 is entered to one of the combinations and the evaluation metrics are recorded. Moreover, the mean and standard deviation of five iterations is calculated. The obtained results of three classifiers (SVM, KNN and RF) are presented in Tables 5, 6 and 7 respectvely. We noted that the results for user 1 dataset shows

**Table 5  Performance metrics of SVM algorithms using ten users dataset.**

| User | OP | | Time | NSF | Acc | F1_score | AUC | Precision | Recall | Specificity | Kappa |
|------|-----|------|------|------|--------|----------|------|-----------|--------|-------------|-------|
| 1 | HHO | mean | 40.32 | 24.20 | 96.13 | 0.96 | 0.96 | 0.93 | 1.00 | 1.00 | 0.92 |
| | | std | 11.09 | 3.11 | 0.56 | 0.01 | 0.01 | 0.01 | 0.00 | 0.00 | 0.01 |
| | WOA | mean | 14.89 | 17.80 | 96.80 | 0.97 | 0.97 | 0.94 | 0.99 | 0.99 | 0.94 |
| | | std | 2.64 | 10.92 | 0.56 | 0.01 | 0.01 | 0.02 | 0.01 | 0.01 | 0.01 |
| 6 | HHO | mean | 58.96 | 18.20 | 97.05 | 0.97 | 0.97 | 0.94 | 1.00 | 1.00 | 0.94 |
| | | std | 9.64 | 5.54 | 0.48 | 0.00 | 0.00 | 0.01 | 0.00 | 0.00 | 0.01 |
| | WOA | mean | 12.08 | 12.60 | 97.05 | 0.97 | 0.97 | 0.95 | 0.99 | 0.99 | 0.94 |
| | | std | 0.97 | 4.93 | 0.48 | 0.00 | 0.00 | 0.02 | 0.02 | 0.02 | 0.01 |
| 8 | HHO | mean | 26.17 | 8.40 | 100.00 | 1.00 | 1.00 | 1.00 | 1.00 | 1.00 | 1.00 |
| | | std | 3.93 | 2.97 | 0.00 | 0.00 | 0.00 | 0.00 | 0.00 | 0.00 | 0.00 |
| | WOA | mean | 14.73 | 7.60 | 100.00 | 1.00 | 1.00 | 1.00 | 1.00 | 1.00 | 1.00 |
| | | std | 4.41 | 2.41 | 0.00 | 0.00 | 0.00 | 0.00 | 0.00 | 0.00 | 0.00 |
| 9 | HHO | mean | 11.14 | 8.20 | 100.00 | 1.00 | 1.00 | 1.00 | 1.00 | 1.00 | 1.00 |
| | | std | 2.11 | 6.69 | 0.00 | 0.00 | 0.00 | 0.00 | 0.00 | 0.00 | 0.00 |
| | WOA | mean | 20.91 | 7.80 | 100.00 | 1.00 | 1.00 | 1.00 | 1.00 | 1.00 | 1.00 |
| | | std | 8.19 | 4.60 | 0.00 | 0.00 | 0.00 | 0.00 | 0.00 | 0.00 | 0.00 |
| 11 | HHO | mean | 22.83 | 9.80 | 95.48 | 0.95 | 0.96 | 0.92 | 1.00 | 1.00 | 0.91 |
| | | std | 2.68 | 7.29 | 1.43 | 0.01 | 0.01 | 0.03 | 0.01 | 0.01 | 0.03 |
| | WOA | mean | 28.08 | 10.60 | 95.12 | 0.95 | 0.95 | 0.91 | 1.00 | 1.00 | 0.90 |
| | | std | 32.12 | 8.32 | 0.78 | 0.01 | 0.01 | 0.01 | 0.00 | 0.00 | 0.02 |
| 12 | HHO | mean | 49.43 | 20.40 | 96.12 | 0.96 | 0.96 | 0.95 | 0.98 | 0.98 | 0.92 |
| | | std | 14.34 | 9.89 | 1.33 | 0.01 | 0.01 | 0.02 | 0.02 | 0.02 | 0.03 |
| | WOA | mean | 64.26 | 22.80 | 96.42 | 0.96 | 0.96 | 0.95 | 0.98 | 0.98 | 0.93 |
| | | std | 55.36 | 11.21 | 1.11 | 0.01 | 0.01 | 0.02 | 0.01 | 0.01 | 0.02 |
| 14 | HHO | mean | 90.68 | 25.80 | 99.89 | 1.00 | 1.00 | 1.00 | 1.00 | 1.00 | 1.00 |
| | | std | 165.29 | 9.23 | 0.24 | 0.00 | 0.00 | 0.01 | 0.00 | 0.00 | 0.00 |
| | WOA | mean | 14.13 | 17.60 | 100.00 | 1.00 | 1.00 | 1.00 | 1.00 | 1.00 | 1.00 |
| | | std | 4.24 | 5.68 | 0.00 | 0.00 | 0.00 | 0.00 | 0.00 | 0.00 | 0.00 |
| 26 | HHO | mean | 52.13 | 22.80 | 94.05 | 0.94 | 0.94 | 0.90 | 0.99 | 0.99 | 0.88 |
| | | std | 20.12 | 11.30 | 0.72 | 0.01 | 0.01 | 0.01 | 0.01 | 0.01 | 0.01 |
| | WOA | mean | 36.57 | 20.20 | 94.05 | 0.94 | 0.94 | 0.90 | 0.99 | 0.99 | 0.88 |
| | | std | 11.07 | 8.38 | 1.52 | 0.01 | 0.01 | 0.02 | 0.01 | 0.01 | 0.03 |
| 46 | HHO | mean | 33.08 | 13.80 | 99.72 | 1.00 | 1.00 | 0.99 | 1.00 | 1.00 | 0.99 |
| | | std | 19.99 | 5.45 | 0.39 | 0.00 | 0.00 | 0.01 | 0.00 | 0.00 | 0.01 |
| | WOA | mean | 8.43 | 16.60 | 100.00 | 1.00 | 1.00 | 1.00 | 1.00 | 1.00 | 1.00 |
| | | std | 2.63 | 2.97 | 0.00 | 0.00 | 0.00 | 0.00 | 0.00 | 0.00 | 0.00 |
| 49 | HHO | mean | 19.59 | 18.80 | 98.53 | 0.98 | 0.99 | 0.98 | 0.99 | 0.99 | 0.97 |
| | | std | 0.94 | 3.70 | 0.87 | 0.01 | 0.01 | 0.01 | 0.01 | 0.01 | 0.02 |
| | WOA | mean | 33.75 | 22.00 | 99.60 | 1.00 | 1.00 | 1.00 | 0.99 | 0.99 | 0.99 |
| | | std | 3.09 | 8.60 | 0.37 | 0.00 | 0.00 | 0.01 | 0.01 | 0.01 | 0.01 |

**Table 6  Performance metrics of KNN algorithms using ten users dataset.**

| User | OP | | Time | NSF | Acc | F1_score | AUC | Precision | Recall | Specificity | Kappa |
|------|-----|------|--------|-------|--------|----------|------|-----------|--------|-------------|-------|
| 1 | HHO | mean | 116.50 | 10.80 | 97.60 | 0.98 | 0.98 | 0.95 | 1.00 | 1.00 | 0.95 |
| | | std | 97.38 | 13.65 | 1.01 | 0.01 | 0.01 | 0.02 | 0.00 | 0.00 | 0.02 |
| | WOA | mean | 56.17 | 18.60 | 97.47 | 0.97 | 0.98 | 0.95 | 1.00 | 1.00 | 0.95 |
| | | std | 32.53 | 16.36 | 0.30 | 0.00 | 0.00 | 0.01 | 0.01 | 0.01 | 0.01 |
| 6 | HHO | mean | 146.35 | 22.20 | 98.75 | 0.99 | 0.99 | 0.98 | 1.00 | 1.00 | 0.97 |
| | | std | 80.88 | 11.03 | 0.74 | 0.01 | 0.01 | 0.01 | 0.01 | 0.01 | 0.01 |
| | | mean | 50.60 | 24.60 | 99.09 | 0.99 | 0.99 | 0.98 | 1.00 | 1.00 | 0.98 |
| | WOA | std | 21.87 | 8.05 | 0.51 | 0.01 | 0.00 | 0.01 | 0.00 | 0.00 | 0.01 |
| 8 | HHO | mean | 20.12 | 3.60 | 100.00 | 1.00 | 1.00 | 1.00 | 1.00 | 1.00 | 1.00 |
| | | std | 9.61 | 2.51 | 0.00 | 0.00 | 0.00 | 0.00 | 0.00 | 0.00 | 0.00 |
| | WOA | mean | 18.37 | 4.00 | 100.00 | 1.00 | 1.00 | 1.00 | 1.00 | 1.00 | 1.00 |
| | | std | 4.34 | 1.87 | 0.00 | 0.00 | 0.00 | 0.00 | 0.00 | 0.00 | 0.00 |
| 9 | HHO | mean | 37.81 | 5.80 | 100.00 | 1.00 | 1.00 | 1.00 | 1.00 | 1.00 | 1.00 |
| | | std | 41.71 | 8.67 | 0.00 | 0.00 | 0.00 | 0.00 | 0.00 | 0.00 | 0.00 |
| | WOA | mean | 24.53 | 3.00 | 100.00 | 1.00 | 1.00 | 1.00 | 1.00 | 1.00 | 1.00 |
| | | std | 9.79 | 4.47 | 0.00 | 0.00 | 0.00 | 0.00 | 0.00 | 0.00 | 0.00 |
| 11 | HHO | mean | 143.89 | 13.80 | 96.55 | 0.96 | 0.97 | 0.93 | 1.00 | 1.00 | 0.93 |
| | | std | 80.31 | 7.79 | 0.98 | 0.01 | 0.01 | 0.02 | 0.01 | 0.01 | 0.02 |
| | WOA | mean | 21.41 | 11.20 | 96.90 | 0.97 | 0.97 | 0.94 | 1.00 | 1.00 | 0.94 |
| | | std | 21.04 | 10.18 | 1.36 | 0.01 | 0.01 | 0.02 | 0.01 | 0.01 | 0.03 |
| 12 | HHO | mean | 81.39 | 8.20 | 96.87 | 0.97 | 0.97 | 0.95 | 0.98 | 0.98 | 0.94 |
| | | std | 77.52 | 11.37 | 1.53 | 0.02 | 0.02 | 0.03 | 0.03 | 0.03 | 0.03 |
| | WOA | mean | 47.77 | 6.60 | 96.12 | 0.96 | 0.96 | 0.95 | 0.97 | 0.97 | 0.92 |
| | | std | 48.95 | 9.13 | 0.33 | 0.00 | 0.00 | 0.03 | 0.04 | 0.04 | 0.01 |
| 14 | HHO | mean | 127.96 | 28.40 | 99.78 | 1.00 | 1.00 | 1.00 | 1.00 | 1.00 | 1.00 |
| | | std | 30.69 | 5.37 | 0.30 | 0.00 | 0.00 | 0.01 | 0.00 | 0.00 | 0.01 |
| | WOA | mean | 58.06 | 21.60 | 99.78 | 1.00 | 1.00 | 1.00 | 1.00 | 1.00 | 1.00 |
| | | std | 23.61 | 7.23 | 0.49 | 0.01 | 0.00 | 0.01 | 0.00 | 0.00 | 0.01 |
| 26 | HHO | mean | 75.24 | 17.60 | 95.95 | 0.96 | 0.96 | 0.92 | 1.00 | 1.00 | 0.92 |
| | | std | 24.06 | 11.67 | 0.72 | 0.01 | 0.01 | 0.01 | 0.01 | 0.01 | 0.01 |
| | WOA | mean | 178.57 | 20.20 | 96.33 | 0.96 | 0.96 | 0.94 | 0.98 | 0.98 | 0.93 |
| | | std | 155.95 | 4.44 | 0.53 | 0.00 | 0.00 | 0.02 | 0.01 | 0.01 | 0.01 |
| 46 | HHO | mean | 152.98 | 17.60 | 99.72 | 1.00 | 1.00 | 0.99 | 1.00 | 1.00 | 0.99 |
| | | std | 66.12 | 5.08 | 0.39 | 0.00 | 0.00 | 0.01 | 0.00 | 0.00 | 0.01 |
| | WOA | mean | 68.17 | 20.20 | 100.00 | 1.00 | 1.00 | 1.00 | 1.00 | 1.00 | 1.00 |
| | | std | 20.73 | 6.76 | 0.00 | 0.00 | 0.00 | 0.00 | 0.00 | 0.00 | 0.00 |
| 49 | HHO | mean | 182.38 | 20.00 | 99.47 | 0.99 | 0.99 | 0.99 | 1.00 | 1.00 | 0.99 |
| | | std | 77.80 | 7.78 | 0.56 | 0.01 | 0.01 | 0.01 | 0.01 | 0.01 | 0.01 |
| | WOA | mean | 114.35 | 27.20 | 99.73 | 1.00 | 1.00 | 1.00 | 1.00 | 1.00 | 0.99 |
| | | std | 19.77 | 3.03 | 0.37 | 0.00 | 0.00 | 0.01 | 0.01 | 0.01 | 0.01 |

**Table 7  Performance metrics of RF algorithms using ten users dataset.**

| User | OP | | Time | NSF | Acc | F1_score | AUC | Precision | Recall | Specificity | Kappa |
|---|---|---|---|---|---|---|---|---|---|---|---|
| 1 | HHO | mean | 1755.65 | 7.20 | 99.33 | 0.99 | 0.99 | 0.99 | 1.00 | 1.00 | 0.99 |
| | | std | 1272.71 | 3.70 | 0.47 | 0.00 | 0.00 | 0.01 | 0.01 | 0.01 | 0.01 |
| | WOA | mean | 997.40 | 7.80 | 99.60 | 1.00 | 1.00 | 0.99 | 1.00 | 1.00 | 0.99 |
| | | std | 600.96 | 4.27 | 0.37 | 0.00 | 0.00 | 0.01 | 0.01 | 0.01 | 0.01 |
| 6 | HHO | mean | 870.45 | 17.40 | 99.66 | 1.00 | 1.00 | 1.00 | 1.00 | 1.00 | 0.99 |
| | | std | 455.22 | 7.27 | 0.31 | 0.00 | 0.00 | 0.01 | 0.01 | 0.01 | 0.01 |
| | WOA | mean | 460.59 | 13.40 | 99.89 | 1.00 | 1.00 | 1.00 | 1.00 | 1.00 | 1.00 |
| | | std | 407.06 | 1.14 | 0.25 | 0.00 | 0.00 | 0.01 | 0.00 | 0.00 | 0.01 |
| 8 | HHO | mean | 305.62 | 4.00 | 99.52 | 1.00 | 1.00 | 0.99 | 1.00 | 1.00 | 0.99 |
| | | std | 50.73 | 4.80 | 1.06 | 0.01 | 0.01 | 0.01 | 0.01 | 0.01 | 0.02 |
| | WOA | mean | 737.74 | 1.80 | 100.00 | 1.00 | 1.00 | 1.00 | 1.00 | 1.00 | 1.00 |
| | | std | 884.06 | 1.79 | 0.00 | 0.00 | 0.00 | 0.00 | 0.00 | 0.00 | 0.00 |
| 9 | HHO | mean | 652.78 | 2.80 | 100.00 | 1.00 | 1.00 | 1.00 | 1.00 | 1.00 | 1.00 |
| | | std | 589.31 | 1.79 | 0.00 | 0.00 | 0.00 | 0.00 | 0.00 | 0.00 | 0.00 |
| | WOA | mean | 798.26 | 1.00 | 100.00 | 1.00 | 1.00 | 1.00 | 1.00 | 1.00 | 1.00 |
| | | std | 1047.19 | 0.00 | 0.00 | 0.00 | 0.00 | 0.00 | 0.00 | 0.00 | 0.00 |
| 11 | HHO | mean | 756.24 | 8.20 | 99.76 | 1.00 | 1.00 | 1.00 | 1.00 | 1.00 | 1.00 |
| | | std | 188.69 | 4.82 | 0.33 | 0.00 | 0.00 | 0.01 | 0.00 | 0.00 | 0.01 |
| | WOA | mean | 505.10 | 7.80 | 99.76 | 1.00 | 1.00 | 1.00 | 1.00 | 1.00 | 1.00 |
| | | std | 51.30 | 4.97 | 0.33 | 0.00 | 0.00 | 0.01 | 0.00 | 0.00 | 0.01 |
| 12 | HHO | mean | 820.00 | 8.20 | 99.25 | 0.99 | 0.99 | 0.99 | 1.00 | 1.00 | 0.99 |
| | | std | 180.44 | 3.42 | 1.06 | 0.01 | 0.01 | 0.02 | 0.01 | 0.01 | 0.02 |
| | WOA | mean | 1151.25 | 8.20 | 98.66 | 0.99 | 0.99 | 0.99 | 0.98 | 0.98 | 0.97 |
| | | std | 801.55 | 2.49 | 0.82 | 0.01 | 0.01 | 0.01 | 0.01 | 0.01 | 0.02 |
| 14 | HHO | mean | 450.89 | 6.00 | 99.89 | 1.00 | 1.00 | 1.00 | 1.00 | 1.00 | 1.00 |
| | | std | 89.42 | 1.87 | 0.24 | 0.00 | 0.00 | 0.01 | 0.00 | 0.00 | 0.00 |
| | WOA | mean | 261.75 | 4.80 | 99.89 | 1.00 | 1.00 | 1.00 | 1.00 | 1.00 | 1.00 |
| | | std | 164.57 | 0.84 | 0.24 | 0.00 | 0.00 | 0.00 | 0.01 | 0.01 | 0.00 |
| 26 | HHO | mean | 506.44 | 10.80 | 99.75 | 1.00 | 1.00 | 0.99 | 1.00 | 1.00 | 0.99 |
| | | std | 342.97 | 6.30 | 0.35 | 0.00 | 0.00 | 0.01 | 0.00 | 0.00 | 0.01 |
| | WOA | mean | 2001.47 | 7.60 | 99.49 | 0.99 | 1.00 | 0.99 | 1.00 | 1.00 | 0.99 |
| | | std | 2052.32 | 3.29 | 0.53 | 0.01 | 0.01 | 0.01 | 0.00 | 0.00 | 0.01 |
| 46 | HHO | mean | 865.38 | 10.40 | 99.44 | 0.99 | 0.99 | 0.99 | 1.00 | 1.00 | 0.99 |
| | | std | 891.30 | 5.41 | 0.31 | 0.00 | 0.00 | 0.01 | 0.00 | 0.00 | 0.01 |
| | WOA | mean | 191.26 | 6.60 | 99.44 | 0.99 | 0.99 | 0.99 | 1.00 | 1.00 | 0.99 |
| | | std | 14.08 | 3.05 | 0.92 | 0.01 | 0.01 | 0.02 | 0.00 | 0.00 | 0.02 |
| 49 | HHO | mean | 497.92 | 6.60 | 99.47 | 0.99 | 0.99 | 0.99 | 1.00 | 1.00 | 0.99 |
| | | std | 48.51 | 2.07 | 0.30 | 0.00 | 0.00 | 0.01 | 0.01 | 0.01 | 0.01 |
| | WOA | mean | 303.64 | 5.40 | 99.47 | 0.99 | 0.99 | 0.99 | 1.00 | 1.00 | 0.99 |
| | | std | 63.78 | 1.95 | 0.87 | 0.01 | 0.01 | 0.02 | 0.00 | 0.00 | 0.02 |

that the combination of WOA-RF surpass the other algorithms in terms of performance metrics. The WOA-RF achieved an average accuracy of 99.60 ± 0.37%. The second metric compares the number of features selected by the two optimization algorithms (HHO and WOA) that able to achieve high performance. HHO-RF chose the fewest number of features in average 7.20 ± 3.70 while attaining an accuracy rate of (99.33 ± 0.47%). Finally, WOA-SVM is faster than other algorithms, finishing 100 generations in a mere 14.89 ± 2.64 s. Consequently, WOA-RF demonstrated the best accuracy performance, HHO-RF excelled in feature reduction, and WOA-SVM showed superior convergence speed.

The results of the experiment for user 6 shows that the combination of WOA-RF surpass the other algorithms in terms of performance metrics. The WOA-RF achieved in average 99.89 ± 0.25% accuracy. The second metric compares the number of features selected by two optimization algorithms (HHO and WOA) for use in its classification step, that each combination was able to use to attain its high performance. WOA-SVM chose the fewest number of features in average 12.60 ± 4.93 while attaining the accuracy rate of (97.05 ± 0.48%). Also, WOA-SVM is faster than other algorithms, finishing 100 generations in a mere 12.08 ± 0.97 s. As a result, we can say that, WOA-RF performs the best in terms of accuracy. WOA-SVM performs the best in both of feature reduction and convergence speed.

The results of the experiment for the user 8 show that all combinations achieved an accuracy of 100% except for HHO-RF which achieved 99.52% ± 1.06. The second metric compares the number of features selected by two optimization algorithms (HHO and WOA). The obtained results demonstrate that, on average, the combination of WOA-RF chose the fewest number of features: 1.80 ± 1.8 features. Finally, WOA-SVM is faster than other algorithms, finishing 100 generations in a mere 14.73 ± 4.41 s. As a result, we can say that, all most combinations act better in terms of accuracy. WOA-RF performs the best in terms of feature reduction, while WOA-SVM performs the best in terms of convergence speed.

## DISCUSSIONS

To compare the performance of different combinations of HHO and WOA combined with SVM, KNN, and RF in solving the legitimacy problem for 10 users, each user run as separated experiment. we conducted cumulative analysis. Each combination that achieved the highest performance was assigned a point of 1, and then we counted the number of points gained by each combination for the three different problems (accuracy, number of feature reduction, and speed). The results are presented in Fig. 3 From the analysis, we can conclude that WOA-RF performs the best in terms of accuracy and feature reduction. Additionally, WOA-SVM demonstrates the best convergence speed for most of the users. The results of our experiments showed consistency with the comprehensive study conducted by *Alwajih et al. (2022)* comparing WOA and HHO for feature selection problems where it was found that WOA has performed better than HHO in these problems. It is important to note that there are a few limitations inherited from the RHU touch mobile keystroke dataset used in the study. These include a small sample size (51 individuals),

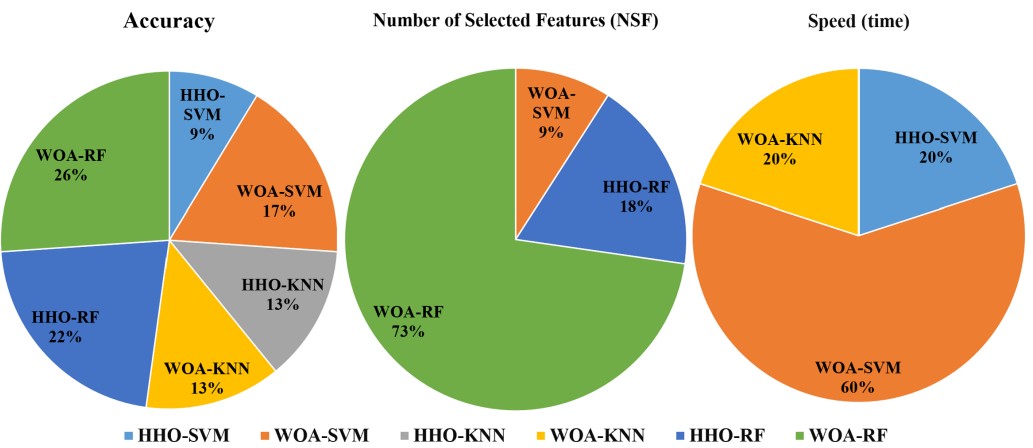

**Figure 3** Commutative comparison of the different combinations in terms of accuracy, NSF and speed.

not covering all age groups, a homogeneous task of entering a single password, and limited demographic details. These constraints may impact the generalizability and representativeness of findings in studying typing behavior for mobile authentication.

## CONCLUSION

The study aims to address computing constraints in smartphones, introducing an efficient user authentication system and addressing public dataset size issues. CTGAN is employed for generating synthetic data from real data. Subsequently, a novel implicit authentication model is presented for smartphone users identification. The proposed model (ONE3A) is utilizes bio-inspred algorithms (WOA and HHO) as feature selection method, combined with one of three classifiers: SVM, KNN and RF. The model (ONE3A) is mirroring real authentication scenarios. The model is evaluated using a publicly available dataset, demonstrating: (1) GAN-generated data quality with scores of 91.84%, 94.47%, and 89.22% for quality score, column shapes, and column pair trends, respectively; (2) WOA outperforming HHO in feature selection for touch-based smartphone authentication; and (3) The combination of WOA-RF is the most effective among various combinations, achieving an average accuracy of 99.62% $\pm$ 0.40% for ten users, with an average reduction rate of 87.85%. In the future, it is aimed to build a bigger study covering data from various age groups with wider demographic details.

### Funding

This work was funded by the University of Jeddah, Jeddah, Saudi Arabia, under grant No. (UJ-20-096-DR). The funders had no role in study design, data collection and analysis, decision to publish, or preparation of the manuscript.

## Grant Disclosures

The following grant information was disclosed by the authors:
The University of Jeddah, Jeddah, Saudi Arabia: UJ-20-096-DR.

## Competing Interests

The authors declare there are no competing interests.

## Author Contributions

- Mohamed Meselhy Eltoukhy conceived and designed the experiments, performed the experiments, analyzed the data, performed the computation work, authored or reviewed drafts of the article, and approved the final draft.
- Tarek Gaber conceived and designed the experiments, performed the experiments, analyzed the data, prepared figures and/or tables, and approved the final draft.
- Abdulwahab Ali Almazroi conceived and designed the experiments, analyzed the data, authored or reviewed drafts of the article, and approved the final draft.
- Marwa F. Mohamed conceived and designed the experiments, performed the experiments, analyzed the data, performed the computation work, prepared figures and/or tables, and approved the final draft.

## Data Availability

   The dataset and code are available in the Supplemental File.

## Supplemental Information

Supplemental information for this article can be found online at http://dx.doi.org/10.7717/peerj-cs.2001#supplemental-information.

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
