# Peer review of "ONE3A: one-against-all authentication model for smartphone using GAN network and optimization techniques"

_PeerJ Computer Science, doi:10.7717/peerj-cs.2001_

## Round 0.1 · original submission · Major Revisions

The authors must accurately revise the manuscript.

**Language Note:** The review process has identified that the English language must be improved. PeerJ can provide language editing services - please contact us at [email protected] for pricing (be sure to provide your manuscript number and title). Alternatively, you should make your own arrangements to improve the language quality and provide details in your response letter. – PeerJ Staff

Reviewer 1 ·

Basic reporting

The work does not seem current to me, in the introduction there is only one work from 2021 and several that are more than 5 years old, what is newly presented?

Experimental design

In the related works section it is necessary to define the research problem, and the problems solved based on the researched literature. A comparative table is necessary, current references also reinforce the news (include recently published works: 2022, 2023, and 2024).

Validity of the findings

It is not clear how the presented results were obtained, or what criteria and metrics were used. Algorithms and functions need more details so that the reader can understand how they arrived at the results.

Additional comments

It is not clear what is new, solved research problems, and scientific contributions.
I suggest that authors update bibliographic references, and define the criteria and parameters used for implementation, tests, and results obtained. If possible, provide readers with an online repository so that the results obtained can be used in other scenarios and applications in the future.

·

Basic reporting

This paper studied on an implicit authentication method for smartphones. Conditional Tabular Generative
Adversarial Network was used to generate synthetic data from actual touch behavior data. In addition, this work used a optimization algorithm called whale optimization algorithm to improve performance of the authentication. While this work offers valuable contributions, addressing a few areas of improvement prior to publication would further strengthen its overall impact.

1.The abbreviation 'WOA-RF' is introduced in the abstract, but the 'RF' component should be defined on its first occurrence prior to subsequent usage of the shortened form.

2.Ensuring the final manuscript is free of minor errors would further strengthen its quality. It is advised that the authors thoroughly proofread the current draft to identify and correct all linguistic oversights, misspellings, formatting inconsistencies.

3.The authors have an opportunity to further enrich the Related Works section by showing previous state-of-the-art efforts through a summary table.

4.To optimize the readability of Figure 2, consider enlarging the text elements overlaid on the pie charts. At their current scaling, the terminology and percentage details along the slices appear slightly undersized.

Experimental design

1) To understand data, the authors should show some sample of touch mobile keystroke data used in this work.

2) What is the reason to select three machine learning algorithms (SVM, KNN, RF). Please explain.

Validity of the findings

1) Expanding the analytical discussion of the experimental results to cover why the proposed WOA approach outperforms the HHO could provide valuable insight.

2) Expanding upon the current limitations of the presented methodology would further strengthen the discourse and help guide constructive future work. Please discuss and provide beneficial future works.

---

## Round 0.2 · accepted · Accept

Based on the reviewers' comments, and the corrections performed by the authors. The authors accurately revised the manuscript, and it can be accepted.

·

Basic reporting

'no comment'

Experimental design

'no comment'

Validity of the findings

'no comment'

Additional comments

'no comment'

·

Basic reporting

The research explores various wrapper feature selection strategies and classifiers to improve authentication accuracy considering smartphone limitations such as hardware constraints, battery life, and memory size.

The article is written in English and uses clear, concise, technically correct text. The article complies with professional standards of courtesy and expression.
Literature references, adequate field history/context provided.
Professional article structure, figures, tables are appropriate. Data has been shared.
In the study, all results related to the hypothesis are given.
The results of the study were tried to be expressed clearly with tables.

Experimental design

In the second part of the study, detailed information about related works is given. It is understandable to present related works with a table
The research question could have been defined more clearly. The research question could be supported with references.
In the third part of the study, the model is explained in detail and supported by figures.
In the 4th section, experimental results are given and the discussion section is carried out.
The experiments are well-designed and described, but the complexity inherent to the subject is not negligible.

Validity of the findings

The validity of the findings is proven. In the study, the data used for the Results are acceptable and ready for use.
The results are expressed appropriately. The data in the tables and figures have been interpreted correctly

Additional comments

in "Respond to the comments of Reviewers" paper, The authors responded and edited the referees' suggestions accordingly.